# Equitable COVID-19 Vaccination for Hispanics in the United States: A Success Story from California Border Communities

**DOI:** 10.3390/ijerph19010535

**Published:** 2022-01-04

**Authors:** Maria Elena Martinez, Jesse N. Nodora, Corinne McDaniels-Davidson, Noe C. Crespo, Amir Adolphe Edward

**Affiliations:** 1Herbert Wertheim School of Public Health and Human Longevity Science, University of California San Diego, La Jolla, CA 92093, USA; jnodora@health.ucsd.edu; 2Moores Cancer Center, University of California San Diego, La Jolla, CA 92093, USA; 3Altman Clinical and Translational Research Institute, University of California San Diego, La Jolla, CA 92037, USA; 4Institute for Public Health, School of Public Health, San Diego State University, San Diego, CA 92020, USA; cmcdaniels@sdsu.edu; 5School of Public Health, San Diego State University, San Diego, CA 92182, USA; ncrespo@sdsu.edu; 6El Centro Regional Medical Center, El Centro, CA 92243, USA; Adolphe.Edward@ecrmc.org

**Keywords:** 2019 novel coronavirus disease, vaccine equity, cancer prevention, health disparities

## Abstract

The ongoing 2019 novel coronavirus disease (COVID-19) pandemic continues to impact the health of individuals worldwide, including causing pauses in lifesaving cancer screening and prevention measures. From time to time, elective medical procedures, such as those used for cancer screening and early detection, were deferred due to concerns regarding the spread of the infection. The short- and long-term consequences of these temporary measures are concerning, particularly for medically underserved populations, who already experience inequities and disparities related to timely cancer care. Clearly, the way out of this pandemic is by increasing COVID-19 vaccination rates and doing so in an equitable manner so that communities most affected receive preferential access and administration. In this article, we provide a perspective on vaccine equity by featuring the experience of the California Hispanic community, who has been disproportionately impacted by the pandemic. We first compared vaccination rates in two United States–Mexico border counties in California (San Diego County and Imperial County) to counties elsewhere in California with a similar Hispanic population size. We show that the border counties have substantially lower unvaccinated proportions of Hispanics compared to other counties. We next looked at county vaccination rates according to the California Healthy Places Index, a health equity metric and found that San Diego and Imperial counties achieved more equitable access and distribution than the rest of the state. Finally, we detail strategies implemented to achieve high and equitable vaccination in this border region, including Imperial County, an agricultural region that was California’s epicenter of the COVID-19 crisis at the height of the pandemic. These United States–Mexico border county data show that equitable vaccine access and delivery is possible. Multiple strategies can be used to guide the delivery and access to other public health and cancer preventive services.

With the surge of new variants, the effects of the coronavirus disease 2019 (COVID-19) pandemic continue to resurface, particularly affecting unvaccinated individuals. Currently, a large proportion of eligible United States residents are unvaccinated; Hispanic and Black individuals are less likely to be vaccinated than their white counterparts [1]. Achieving broad protection through vaccination is clearly our exit from this long nightmare, which has caused approximately 680,000 deaths nationwide. The pandemic has also resulted in interruptions in cancer screening, leading to the delayed diagnosis of cancers and shifts towards more advanced cancers [2]. There is evidence that it has also disrupted delivery of the human papilloma virus (HPV) vaccine and may also be changing parents’ acceptance of the vaccine [3].

Vaccine equity, defined as preferential access and administration to those most affected by the pandemic, is an important goal of the Centers for Disease Control and Prevention (CDC). To understand issues related to vaccine equity, it is necessary to examine the racial/ethnic composition of the people vaccinated. Although the CDC does not yet publicly report these state-level data, they are accessible through state health departments. With its nearly 40 million residents and diverse population, California can contribute to the understanding of vaccine equity by characterizing the racial, ethnic, and geographical vaccination landscape. We used the vaccination progress data reported by the state of California in efforts to make COVID-19 vaccination data transparent and accessible to all Californians [4]. The data include a variety of summary statistics as counts and percentages. These state-level data show that as of 20 September 2021, 69.5% of eligible Californians are fully vaccinated and 8.2% are partially vaccinated; however, these rates vary by racial/ethnic group, region, and the California Healthy Places Index, which is an indicator of the social determinants of health for a given geographic area [4].

The experience of the California Hispanic community provides an important perspective on vaccine equity. Hispanics have been disproportionately impacted by the pandemic [5] and they make up approximately 40% of the state’s population. According to the Kaiser Family Foundation report, vaccination delivery for California Hispanics has not matched the burden of infection: 31% of vaccinations have gone to Hispanic residents yet they account for 61% of COVID-19 cases and 47% of deaths in the state [1]. However, the story is not equally somber across all Hispanics in California. One region in the state with a low proportion of unvaccinated Hispanics is the United States–Mexico border, comprised of San Diego and Imperial Counties. Both counties are considered majority-minority regions, meaning that non-Hispanic White individuals are not the majority population. According to the 2021 U.S. census estimates [6], San Diego County has approximately 3.3 million residents and Hispanic individuals make up 34.1% of the population. In this county, 23.4% are foreign-born individuals, 37.6% speak a language other than English at home, and 9.5% live below the poverty line. Imperial County is a more rural region than San Diego County, with approximately 180,000 residents and 85% of these are Hispanic individuals. This county is made up of 30.7% foreign-born residents, 76.5% speak a language other than English at home, and 18.1% live below the poverty line. The proportion of unvaccinated Hispanics is 14.9% in Imperial County and 26.5% in San Diego County, percentages that are substantially lower than the 41.3% statewide percentage of unvaccinated Hispanics [4]. Understanding the reasons for this success could provide important lessons on how to proactively lower the number of unvaccinated persons elsewhere and to successfully rollout other vaccination programs.

Using these publicly available data, which include a low proportion of missing race/ethnicity data (4.6%) [4], we first compared vaccination status among Hispanics in San Diego County and Imperial County to other California counties with a similar Hispanic population size [3]. As shown in Figure 1, the proportion of unvaccinated Hispanics in four counties comparable to Imperial County ranged from 41.3% in San Mateo County to 54.6% in San Joaquin County, which is much higher than the 20.4% in Imperial County. For San Diego County, the four comparison counties ranged from 35.8% unvaccinated Hispanics in Santa Clara County to 53.1% in Orange County, higher than the proportion in San Diego County (26.5%). These data clearly show a much higher vaccination level for Hispanics living in California border counties compared to those living elsewhere.

We next looked at county vaccination rates according to a health equity metric [7] based on the California Healthy Places Index (HPI) [8]. This metric was developed by the Public Health Alliance of Southern California and uses economic, education, healthcare access, and related indicators to derive quartiles ranging from less healthy (quartile 1) to most healthy (quartile 4) community conditions. As shown in Figure 2, in California, individuals living in less healthy communities have lower vaccination rates than those in healthier communities [4], indicating that the state as a whole has not achieved vaccine equity. In San Diego County, delivery was more equitable than the rest of California, where residents living in communities in the lowest HPI quartile had the second highest vaccination rate (65.8%). Specifically, while the percentage point difference between HPI quartiles 1 and 4 for the state of California was 22.9, that for San Diego County was 12.7. In Imperial County, the highest vaccine uptake was in the community with the lowest HPI quartile (75.3%). Conversely, in all comparison counties shown in Figure 1, the lowest vaccination rates were in the lowest HPI, similar to the state overall profile (data not shown). These United States–Mexico border county data show that equitable vaccine access and delivery is possible, particularly among Hispanics.

Achieving equitable vaccine delivery is aspirational but not easily accomplished. In our last inquiry, we investigated strategies that may have led to this success in vaccination in San Diego and Imperial Counties. In San Diego County, the Health and Human Services Agency (HHSA) led the way with a clear equity focus in their T3 (test, trace, treat) program from the start. Some of the HHSA initiatives included: (1) clear and transparent data reporting by race/ethnicity and geography; (2) a massive scale-up of free and accessible testing (and later vaccination) in communities with the greatest need; (3) funding the development and expansion of community health worker (CHW)-led programs, such as contact tracing and communications and outreach, and priority vaccine appointment scheduling for communities most impacted by the pandemic; and (4) the establishment of community sector support, where 13 business, education, service, healthcare, and priority population sectors were established with regular county-staffed meetings to provide communication and guidance relative to COVID-19. In addition to their own initiatives, the HHSA contracted with trusted community-based organizations and with institutions like San Diego State University and the University of California San Diego, which already had active community outreach and engagement efforts through long-standing federally-funded community–academic partnerships. Both institutions were then able to compete for, and receive, a substantial number of additional federally funded grants to further serve marginalized communities, including its large Hispanic community, through innovative testing and vaccination strategies.

Imperial County is an agricultural region that became California’s COVID-19 epicenter at the height of the pandemic. To address this devastation, communities and community stakeholders stepped up in a major way. While some regions in the country followed the Field of Dreams concept of “if you build it, they will come”, Imperial County knew that more had to be done to increase vaccine access. A strong network of clinics, hospitals, and the public health sector alongside nonprofits, local businesses, and other groups within and outside the county, including San Diego, came together to provide vaccinations. This involved setting up resources to take vaccination directly into priority communities and transporting people to vaccination sites: buses took agricultural workers to vaccination sites, malls set up vaccination stations, and public health workers delivered vaccines in remote rural areas. Importantly, a strong advocacy for vaccine availability for all, regardless of residential address or proof of identification was implemented. This advocacy was crucial, given that ~275,000 United States citizens, including 30,000 retired United States military, live within 10 miles across the Mexico border. These individuals routinely cross into the United States for care, including vaccinations. Reducing the barrier of residential status increases accessibility and promotes trust, resulting in wider vaccine coverage. Imperial County health officials acknowledged that the pandemic did not stop at the border and vaccination delivery should be no different. Social media and health education and promotion campaigns conducted primarily in Spanish created a trusted comfort zone for the community.

## Conclusions

In the end, this is a model story about communities hit hard by the pandemic who overcame many of the barriers other Hispanic communities faced. Vaccines were widely accessible in trusted and convenient locations, all while promoting vaccination through language and culturally tailored health education and promotion. Both counties provided their Hispanic residents equitable and unbiased vaccination, clearly demonstrating that their lives are worth saving. Therein lies an important lesson of true vaccine equity. Perhaps these strategies can be used to guide the delivery and access to other public health and cancer preventive services, such as human papilloma virus vaccination and cancer screening. We look forward to future reports that take into account contextual factors, including additional social determinants of health measures, to explore this topic in more detail.

## Figures and Tables

**Figure 1 ijerph-19-00535-f001:**
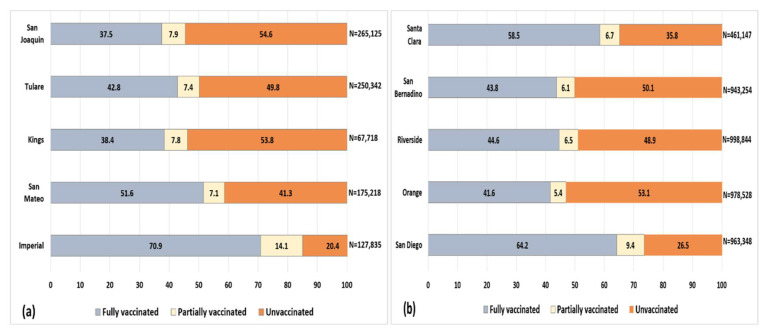
Vaccination status (%) among Hispanics in Imperial County and four comparison counties (**a**) and for San Diego County and four comparison counties (**b**) in California. *n* = number of Hispanic residents in the county.

**Figure 2 ijerph-19-00535-f002:**
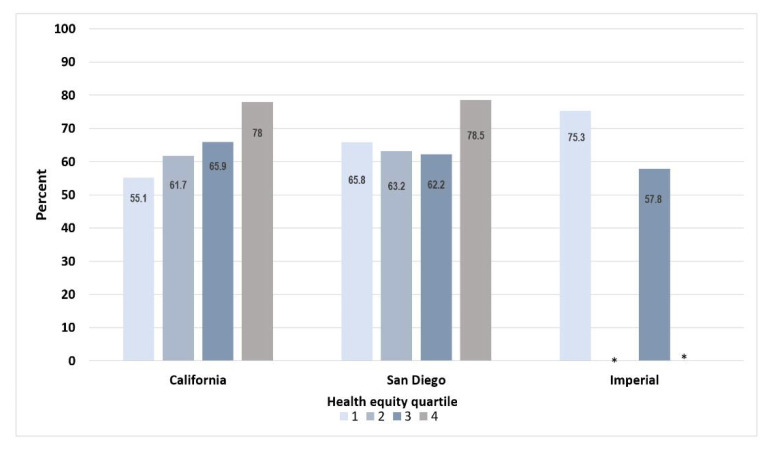
Percent fully vaccinated individuals according to the health equity quartile, which uses the California Healthy Places Index. Regions range from less healthy community conditions (quartile 1) to more healthy community conditions (quartile 4). * Indicates insufficient data.

## Data Availability

Publicly available datasets were analyzed in this study. These data can be found here: https://covid19.ca.gov/vaccination-progress-data/#overview (accessed on 25 September 2021).

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
