# Peer review of "Equitable COVID-19 Vaccination for Hispanics in the United States: A Success Story from California Border Communities"

_ijerph, 2022, doi:10.3390/ijerph19010535_

Round 1

Reviewer 2 Report

Thank you for asking me to review this article. Ensuring equal access to vaccination during the current pandemic, paying particular attention to vulnerable people who have difficulty accessing health services is essential, especially during a pandemic emergency. Reflection on these issues, in fact, offers food for thought for the implementation of strategies aimed at avoiding discrimination due to material factors, such as logistical and administrative aspects, vaccination costs, socio-cultural conditions, any states of deprivation etc.

However, despite the social relevance of the topic in question, it is believed that, given the organization of the contents and the description of the same, the manuscript cannot be published in its current form. As described, in fact, no merit is given to the importance of the subject examined.

In particular, the objectives that guided the research question in the authors' commentary and the hypotheses that the research intended to verify are examined in a dispersive and not very intuitive way for the reader, especially in relation to the reference context. For example, the contents present in the abstract do not allow the reader to have a clear overview of the phenomenon being analyzed, in fact, what is described in the introduction with reference to the impact of the pandemic on prevention measures such as screening with consequent late diagnosis or accessibility to health services especially in deprived cohorts, is not consistent with the title chosen by the authors and should act as a corollary to the problem examined (fairness of anti-COVID-19 vaccination and possible influence of deprivation factors in the use of health services) in the discussion section and / or in the concluding remarks.Furthermore, in the premises, the reference context of the populations examined should be described with a dedicated “study setting” section in order to allow the reader to understand the reference to the context. In the abstract the authors cite… “In this article, we provide a perspective on vaccine equity by featuring the experience of the California Hispanic community, who has been disproportionately impacted by the pandemic. We first compared vaccination rates in the two United States-Mexico border counties of San Diego County and Imperial County to counties elsewhere in California with similar Hispanic population size .. " brief description useful to understand the reasons underlying the research question in the considered cohort. It is suggested to add a paragraph relating to the study environment describing not only the demographic characteristics of the counties considered, but also the socio-health ones that could better define the problem of inequality and open food for thought on possible intervention strategies.
Furthermore, given the importance of the topic under consideration and the different interpretations of the social phenomenon that underlies an unequal vaccination opportunity in some cohorts of individuals, the contents described should be deepened, verified, supported by an in-depth bibliographic investigation and analyzed through different interpretations. In this regard it is suggested to consult the manuscripts i) 10.1016 / j.puhe.2005.01.015; ii) doi: 10.1186 / s12942-015-0004-x; iii) doi: 10.15167 / 2421-4248 / jpmh2018.59.4s2.1077; iv) doi: 10.1186 / 1476-072X-6-27 and many others described in the literature on the same topic.
The presentation of the manuscript should be made more precise and engaging, with a more accurate order of topics including the methodology chosen and the description of the results that make reading difficult. It would also be interesting to broaden the empirical value of the results with reference to future prospects.
Therefore, authors are advised to dig deeper into the content and rearrange their description so that they can resubmit the article for review. Taking into account these considerations, in fact, precise insights on this topic could represent an important contribution to the scientific literature.

Author Response

Comment: Thank you for asking me to review this article. Ensuring equal access to vaccination during the current pandemic, paying particular attention to vulnerable people who have difficulty accessing health services is essential, especially during a pandemic emergency. Reflection on these issues, in fact, offers food for thought for the implementation of strategies aimed at avoiding discrimination due to material factors, such as logistical and administrative aspects, vaccination costs, socio-cultural conditions, any states of deprivation etc. However, despite the social relevance of the topic in question, it is believed that, given the organization of the contents and the description of the same, the manuscript cannot be published in its current form. As described, in fact, no merit is given to the importance of the subject examined.        

In particular, the objectives that guided the research question in the authors' commentary and the hypotheses that the research intended to verify are examined in a dispersive and not very intuitive way for the reader, especially in relation to the reference context. For example, the contents present in the abstract do not allow the reader to have a clear overview of the phenomenon being analyzed, in fact, what is described in the introduction with reference to the impact of the pandemic on prevention measures such as screening with consequent late diagnosis or accessibility to health services especially in deprived cohorts, is not consistent with the title chosen by the authors and should act as a corollary to the problem examined (fairness of anti-COVID-19 vaccination and possible influence of deprivation factors in the use of health services) in the discussion section and / or in the concluding remarks. Furthermore, in the premises, the reference context of the populations examined should be described with a dedicated “study setting” section in order to allow the reader to understand the reference to the context. In the abstract the authors cite… “In this article, we provide a perspective on vaccine equity by featuring the experience of the California Hispanic community, who has been disproportionately impacted by the pandemic. We first compared vaccination rates in the two United States-Mexico border counties of San Diego County and Imperial County to counties elsewhere in California with similar Hispanic population size .. " brief description useful to understand the reasons underlying the research question in the considered cohort. It is suggested to add a paragraph relating to the study environment describing not only the demographic characteristics of the counties considered, but also the socio-health ones that could better define the problem of inequality and open food for thought on possible intervention strategies.

Furthermore, given the importance of the topic under consideration and the different interpretations of the social phenomenon that underlies an unequal vaccination opportunity in some cohorts of individuals, the contents described should be deepened, verified, supported by an in-depth bibliographic investigation and analyzed through different interpretations. In this regard it is suggested to consult the manuscripts i) 10.1016 / j.puhe.2005.01.015; ii) doi: 10.1186 / s12942-015-0004-x; iii) doi: 10.15167 / 2421-4248 / jpmh2018.59.4s2.1077; iv) doi: 10.1186 / 1476-072X-6-27 and many others described in the literature on the same topic.
The presentation of the manuscript should be made more precise and engaging, with a more accurate order of topics including the methodology chosen and the description of the results that make reading difficult. It would also be interesting to broaden the empirical value of the results with reference to future prospects.

Therefore, authors are advised to dig deeper into the content and rearrange their description so that they can resubmit the article for review. Taking into account these considerations, in fact, precise insights on this topic could represent an important contribution to the scientific literature.

Response:  We appreciate the reviewer’s perspective and agree that a “deeper dig” into the content would be valuable. The reviewer makes excellent points and provides very useful ideas for digging deeper into this topic.  However, we are challenged by the limited data available.  We had to rely on publicly-available data from the state of California and we were fortunate to have the data used to construct this Commentary available to the public.  These data are merely summary statistics provided as counts and percentages, which do not permit the exploration beyond what is presented in this Commentary.  As noted in the manuscript draft, at the time of the submission, even the Centers for Disease Control and Prevention did not have similar data for the U.S.; and there are individual states that do not provide these valuable data.  Further, the health equity metric was already derived by the state government agency, with no additional data available to the public, which does not permit further exploration. This is the main reason why we are writing this in the form of a Commentary and not a research manuscript.  As such, we are not able to test a specific hypothesis.  Since the submission of this manuscript, however, the state has begun to make a larger data set available to the public.  We have begun to work with another group of researchers to explore the contextual factors associated with differences in vaccine uptake by race/ethnicity and a variety of census-level neighborhood contextual factors.  In fact, we have a paper under review assessing census-level neighborhood contextual factors in the context of COVID-19 infection rates. As soon as we are able to secure similar data to assess vaccination rates, we will move forward with that manuscript and will consider the valuable recommendations made by the reviewer.

In response to the reviewer’s comments, we have added additional text on the source of the data, lines 54-56, and line 79.  We would like to note that this is a Commentary and as such, not written like a traditional scientific paper.  Lasty, we added a final sentence in the Conclusions section, lines 164-166, related to the excellent recommendations made by this reviewer.   

Reviewer 3 Report

Although this is not a research article, this commentary illustrates the results of a survey aimed at evaluating the factors that may have led to increased access to COVID-19 vaccination for Hispanic communities in US border counties. Unfortunately, the article does not provide sufficient supporting data. The criteria with which some comparisons were made have not been clarified (see comments on the text), there are not enough bibliographic references to support the data reported and the strategies adopted to increase access to vaccination are not illustrated in detail. Attached please find the manuscript with specific comments.

Author Response

Comment:  Although this is not a research article, this commentary illustrates the results of a survey aimed at evaluating the factors that may have led to increased access to COVID-19 vaccination for Hispanic communities in US border counties. Unfortunately, the article does not provide sufficient supporting data. The criteria with which some comparisons were made have not been clarified (see comments on the text), there are not enough bibliographic references to support the data reported and the strategies adopted to increase access to vaccination are not illustrated in detail. Attached please find the manuscript with specific comments.

Response:  We thank the reviewer for the recommendations, which were provided directly into the uploaded pdf manuscript document.  Below are the responses.  Please note that we copied and pasted all comments verbatim, without correction of any grammatical errors.

Comment: There are many definitions of what an equitable access is. "those most affected by the pandemic" means those who are at the highest risk of mortality and morbidity due to COVID-19 or the communities with the higher prevalence and incidence? Please argue and add some references, many important articles have been published on the issue of equitable access for minorities and specifically for hispanci communities, but I can find no specific reference here.

Response:  We appreciate the reviewer’s comment regarding this important topic.  The comment here was placed directly next to the text in the manuscript related to CDC’s definition of vaccine equity, lines 47-49.  While the reviewer asks us to “argue and add some references”, we do not feel it is our responsibility to question the CDC’s definition in this context.  Indeed, there have been many great articles on this topic, including one of our own, which we refer to in the manuscript. 

Comment:  Please provide date for vaccination rates on line 57.

Response:  The date has been included.

Comment:  Please provide an explanation of why different counties were chosen to make the comparison and on which criteria were adopted in the selection of the counties to be compared (population? land area? proportion of non english pseakers? poverty index?)

Response:  We note in the original paragraph related to the comment that the counties are similar in the size of the Hispanic population.  Comparisons beyond this would require more data to be available so we can do a formal adjustment for any differences in the characteristics.  As noted in our response to reviewer number 2, these more detailed data are beginning to be available publicly by the state of California and will be the basis of a future modeling paper.

Comment:  The reviewer is asking the authors to provide references for text made as part of a Commentary in several places in the paragraph prior to the Conclusion statement. 

Response: The text provided in most of the paragraph has not been published as it comprises comments from co-author Dr. Edward, director of El Centro Regional Medical Center, the largest health system in Imperial County.  He has reported this information to the lay media on several occasions.  Beyond that, the context of this paragraph is largely editorial, in line with the Commentary nature of the report. 

Round 2

Reviewer 2 Report

The authors responded to the suggestions proposed in my first review. In my opinion, the structure of a manuscript in the form of a "Commentary" can determine the summary description of broader themes. Nevertheless, it must offer the reader all the tools to understand the point of view of the authors and possibly reflect on ideas of re-examination that emerged after the reading and / or replayability of the research conducted also in other contexts. I therefore suggest that authors approach the subject in writing a manuscript in its "article" form. However, given the authors' observations, I believe that the commentary can be published while advising the authors to pay more attention to the bibliography which I believe is still very sparse given the importance of the topic.

Author Response

Comment: The authors did not make those substantial changes necessary to give scientific soundness to the manuscript. Nothing substantial has been changed.

Response:  We agree with the reviewer regarding the lack of substantial changes.  As we noted in our prior response, we agreed with the reviewer’s perspective on the value of a “deeper dig” into the topic.  However, the deeper dive that the reviewer is calling for relates to his/her/their perspective that our objectives are “guided by the research question” in our commentary.  Again, we would like to clarify that this is not a scientific paper but a Commentary.  To fully explore a hypothesis and research question, much more detailed individual-level or regional-level level data are needed.  That public data that we had available are simply summary statistics, which do not permit us to give the level of “scientific soundness” that the reviewer is requesting.  We therefore call for future work to conduct this much needed work.    

Reviewer 3 Report

The authors did not make those substantial changes necessary to give scientific soundness to the manuscript. Nothing substantial has been changed.

Author Response

Comment: The authors responded to the suggestions proposed in my first review. In my opinion, the structure of a manuscript in the form of a "Commentary" can determine the summary description of broader themes. Nevertheless, it must offer the reader all the tools to understand the point of view of the authors and possibly reflect on ideas of re-examination that emerged after the reading and / or replayability of the research conducted also in other contexts. I therefore suggest that authors approach the subject in writing a manuscript in its "article" form. However, given the authors' observations, I believe that the commentary can be published while advising the authors to pay more attention to the bibliography which I believe is still very sparse given the importance of the topic.

Response:  We thank the reviewer for his/hers/their further comments. We are unclear about the reviewer’s suggestions to write this as an “article” form.  We are also unclear as to what additional bibliography the reviewer would like to have included.  The references that the reviewer requested in the prior review largely pertained to those supporting the data reported, which we included in the revised document.  In our experience, Commentary reports tend to have limited bibliography.  In fact, some journals greatly limit the number of references because these publications are not meant to provide a full bibliography on the topic.